



# Modeling liquid transport in the Earth's mantle as two-phase flow: Effect of an enforced positive porosity on liquid flow and mass conservation

Changyeol Lee[1,*], Nestor G. Cerpa[2,*], Dongwoo Han[1], Ikuko Wada[3]

[1]Department of Earth System Sciences, Yonsei University, Republic of Korea
[2]Geosciences Montpellier, Université de Montpellier, CNRS, Université des Antilles, Place Eugène Bataillon, 34095 Montpellier, France
[3]Department of Earth and Environmental Sciences, University of Minnesota, USA

*Correspondence to*: Changyeol Lee (changyeol.lee@yonsei.ac.kr)
[*]These authors equally contributed to this work.

**Abstract.** Fluid and melt transport in the solid mantle can be modeled as a two-phase flow in which the liquid flow is resisted

by the compaction of the viscously deforming solid mantle. Given the wide impact of the liquid transport on the geodynamical and geochemical evolution of the Earth, the so-called "compaction equations" are more and more incorporated in geodynamical modeling studies. The implementation of these equations requires a regularization technique to handle the porosity singularity in dry mantle. Moreover, it is also common to enforce a positive porosity (liquid fraction) to avoid unphysical negative values of porosity. However, the effects of this "capped" porosity on the liquid transport and mass conservation have not been

quantitatively evaluated. Here, we investigate these effects using a series of 1- and 2-dimensional numerical models using the commercial finite element package COMSOL Multiphysics®. The results of benchmarking experiments against a semi-analytical solution for 1- and 2-D solitary waves illustrate the successful implementation of the compaction equations. We show that the solutions are accurate when the element size is smaller than half of the compaction length. Furthermore, in time-evolving experiments where the solid is stationary (immobile), we show that the mass balance errors are similarly low for both

capped and uncapped experiments (i.e., allowing negative porosity). When Couette flow, convective flow, or subduction corner flow of the solid mantle is assumed, the capped porosity leads to overestimations of the mass of liquid in the model domain and mass flux of liquid across the model boundaries, resulting in intrinsic errors in mass conservation even if high mesh resolution is used. Despite the errors in mass balance, however, the general trends of porosity evolution in the capped experiments are similar to those in the uncapped experiments. Hence, the use of the regularization of the compaction equations

with the enforced positive porosity is reasonable for modeling fluid and melt transport in a deforming mantle.



## 1 Introduction

The fluid and melt within the Earth's mantle, as well as their transport from depth to surface play a key role in the geodynamical and geochemical evolution of our planet. At depth, the presence of small fluid and melt fractions (up to 1-10%) affects the
bulk physical properties of mantle rocks (Mei et al., 2002; Zimmerman and Kohlstedt, 2004; Dohmen and Schmeling, 2021). Such an effect partly influences the vigor of mantle convection (e.g., Ogawa and Nakamura, 1998), potentially assists the localization of deformation (Holtzman et al., 2003; Zimmerman and Kohlstedt, 2004; Katz et al., 2006), and may thus be a key ingredient for the functioning of plate tectonics. At the depths of their generation and through their journey to the surface, fluid and melt extract incompatible and fluid-mobile elements from the mantle rocks, thereby controlling the planetary
differentiation and contributing to the growth of continental crust (Gerya and Meilick, 2011; Jagoutz and Kelemen, 2015). The ascent and eruption of magmas lead to the formation of volcanoes over and between tectonic plates, link the solid Earth evolution to the atmosphere evolution (Lopez et al., 2023).

Because of the wide impact that fluid and melt have on the Earth system, it is crucial to constrain their migration pathways
and spatial distribution. These can be inferred on the basis of geophysical imaging (e.g., magneto-tellurics and seismic tomography). However, such methods lead to interpretations that are often non-unique given the dependence of the observables on multiple factors. Further, they are only present-day static images of a dynamic process. Forward modeling of liquid transport is a tool that can help to quantify the fluid and melt migration and their spatial distribution in the mantle and to study the coupled fluid/melt-mantle dynamics at a geodynamic scale (1-to-100s of kilometer scales).


One of the pioneering studies on liquid (aqueous fluid and melt) transport in the solid Earth was that of McKenzie (1984) (see also Scott and Stevenson, 1984; Fowler, 1985) who derived a two-phase flow theory based on a continuum mechanics for a liquid in a viscously deforming porous solid matrix (mantle rocks). In this theory, a buoyant liquid phase percolates through the solid phase where the liquid viscosity is many orders of magnitude lower than that of the permeable mantle matrix. The
liquid flow follows the Darcy's law but experiences resistance due to the compaction of the solid matrix. Using this theory, liquid flow has been evaluated in various geodynamic settings including mid-ocean ridges (Katz, 2008; Keller and Katz, 2016; Cerpa et al., 2018; Sim et al., 2020; Pusok et al., 2022), subduction zones (Dymkova and Gerya, 2013; Wilson et al., 2014; Cerpa et al., 2017, 2018; Wang et al., 2019; Rees Jones et al., 2018), continental rifts (Schmeling, 2010; Li et al., 2023), and intraplate context (Keller et al., 2013; Dannberg and Heister, 2016).


The mantle away from the vicinity of the plate boundaries is generally thought to be relatively dry except in the specific regions where the presence of volatiles and melts have been suggested, e.g. in the shallow asthenospheric mantle (Chantel et al., 2016; Debayle et al., 2020; Cerpa et al., 2019), and near the 410- and 660-km discontinuities (Bercovici and Karato, 2003). However, in the application of the two-phase flow equations to the mantle, the near zero-porosity limit leads to a singularity, and it is



therefore difficult to handle numerically (Arbogast et al., 2017; Dannberg et al., 2019). Thus, the equations are commonly regularized by imposing a small porosity in the entire model domain (e.g., Wilson et al., 2014; Cerpa et al., 2017). Along with an assumption of small porosity, it is also required to "cap" the porosity field to avoid the development of negative porosity values which naturally arises from the governing equations. However, despite the capped porosity being broadly used in numerical models, its impacts on the liquid transport and mass conservation have not been quantitatively evaluated.


In the present study, we investigate the effect of the regularization with the capped porosity on the liquid transport and mass conservation in the two-phase flow model for the Earth's mantle, using the commercial finite element package COMSOL Multiphysics® (COMSOL, hereafter). COMSOL has been used previously in the context of mantle convection and successfully benchmarked (e.g., Lee, 2013; Yu and Lee, 2018; Trim et al., 2021). COMSOL was used to study the liquid

transport in the mantle wedge of subduction zones, considering a simplified porous flow model without the incorporation of the effect of matrix compaction (e.g., Wada and Behn, 2015; Lee et al., 2021; Kim et al., 2022). Here, we implement the governing equations that account for the compaction of the mantle matrix in COMSOL and validate the implementation by benchmarking the model solution against a semi-analytical solution for 1- and 2-dimensional (1- and 2-D) solitary waves. We then evaluate the effects of a capped porosity on liquid flow and mass conservation by comparing the mass balance between

the capped and uncapped experiments using four different flow fields for the solid matrix: stagnant, Couette flow, convective flow, and subduction corner flow. One of the advantages of COMSOL is that it has the potential to perform the coupling between different physics. Thus, the results of the present study can provide the basis for future applications of COMSOL for coupling the two-phase flow equations with other solid Earth processes, such as chemical reactions and heat transfer by liquids.

**2. Governing equations**

We follow the re-formulation of the physics of two-phase flow in the mantle in which only the solid-state mantle flow (solid flow) influences the porous flow (i.e., one-way coupling) under the small-porosity approximation (e.g., Spiegelman, 1993; Katz et al., 2007; Katz, 2022). Such a formulation has been described in detail by previous studies, which provide the derivation of the non-dimensionalized governing equations for the solid and liquid flow (Wilson et al., 2014; Cerpa et al., 2017). Below,

we briefly describe the equations.

The governing equations for solid flow are the non-dimensionalized incompressible Stokes equations and the heat equation in a non-dimensionalized form (Wilson et al., 2014; Cerpa et al., 2017; Lee et al., 2021):

$$\nabla \cdot \boldsymbol{v}_s = 0 \tag{1}$$



$$\boldsymbol{\nabla} \cdot (2\eta \dot{\boldsymbol{\epsilon}}) \, - \, \boldsymbol{\nabla} p + \frac{h_0^2 \rho_{s_0} \alpha_0 \Delta T g_0}{\eta_0 v_{s_0}} T \boldsymbol{k}_{up} = 0 \qquad (2)$$

$$\frac{\partial T}{\partial t} + \boldsymbol{v}_s \cdot \boldsymbol{\nabla} T = \frac{1}{Pe} \nabla^2 T \qquad (3)$$


where $\boldsymbol{v}_s$ is the solid velocity, $\eta$ is the solid shear viscosity, $\dot{\boldsymbol{\epsilon}}$ is strain rate tensor, $p$ is the dynamic pressure, $h_0$ is the reference length, $\rho_{s_0}$ is the reference solid density, $\alpha_0$ is the thermal expansivity, $\Delta T$ is the temperature difference, $g_0$ is the gravitational acceleration, $\eta_0$ is the reference solid shear viscosity, $v_{s_0}$ is a reference solid velocity, $T$ is the temperature, $\boldsymbol{k}_{up}$ is the unit vector in the direction opposite to the gravity, and $Pe$ is the Peclet number ($h_0 v_{s_0}/\kappa_0$ where $\kappa_0$ is the thermal diffusivity). In

the heat equation, we neglect radiogenic heating and latent heat.

The non-dimensionalized governing equations for liquid flow are:

$$\frac{\partial \phi}{\partial t} + \boldsymbol{v}_s \cdot \boldsymbol{\nabla} \phi - \frac{v_{l_0}}{v_{s_0}} \frac{h_0^2}{\delta_0^2} \frac{P}{\zeta} = \Gamma \qquad (5)$$

$$\frac{h_0^2}{\delta_0^2} \frac{P}{\zeta} - \boldsymbol{\nabla} \cdot \left[ K \left( \boldsymbol{\nabla} P + \frac{\eta_0 v_{s_0}}{\Delta \rho g_0 h_0^2} \boldsymbol{\nabla} p - \boldsymbol{k}_{up} \right) \right] = \frac{v_{s_0}}{v_{l_0}} \frac{\Delta \rho}{\rho_{l_0}} \Gamma \qquad (6)$$

where $\phi$ is the porosity, $P$ is the compaction pressure, $\zeta$ and $K$ are the bulk viscosity and permeability, respectively, and $\Gamma$ is the rate of mass transfer between the solid and liquid phases. The density contrast between the solid and liquid phases is denoted as $\Delta \rho = \rho_{s_0} - \rho_{l_0}$.

The reference compaction length $\delta_0$ is given by

$$\delta_0 = \sqrt{K_0 \phi_0^{n-m} \eta_0} \qquad (7)$$

where $K_0$ is the reference liquid mobility, $\phi_0$ is the reference porosity, $n$ is the permeability exponent, and $m$ is the bulk viscosity exponent (Spiegelman, 1993). The reference liquid velocity $v_{l_0}$ is defined as

$$v_{l_0} = K_0 \phi_0^{n-1} \Delta \rho g_0 \qquad (9)$$

It follows that the non-dimensional relations for the bulk viscosity $\zeta$, the permeability $K$, and the compaction length $\delta$ are, respectively:

$$\zeta = \frac{\eta}{\phi^m} \qquad (10)$$

$$K = \phi^n \qquad (11)$$

$$\delta = \sqrt{\phi^{n-m} \eta} \qquad (12)$$





Given Eq. 10, Eq. 6 may become singular if $\phi \rightarrow 0$. Thus, in most geodynamic applications, where the initial porosity field is
close to zero away from the liquid pathways, some regularizing techniques have been employed to prevent singularity. Here,
we use a regularized bulk viscosity and a regularized permeability, defined as follows: $\tilde{\zeta} = \frac{\eta}{\phi + \phi_\epsilon}$ and $\widetilde{K} = (\phi + \phi_\epsilon)^n$ where
$\phi_\epsilon$ is a user-defined "small" porosity and can be chosen based on the choice of a minimum compaction length $\delta_\epsilon$ following
the relationship $\phi_\epsilon = (\delta_\epsilon^2/\eta)^{\frac{1}{n-m}}$ (Wilson et al., 2014). Note that the regularized permeability is only applied to the term

related to the gradients of the compaction pressure in Eq. 6. Finally, a negative porosity ($\phi < 0$) is physically unrealistic and
thus a non-negative porosity field is commonly imposed as a constraint when solving the equations (e.g., Wilson et al., 2014;
Cerpa et al., 2017). To evaluate the impact such a treatment of the porosity field on liquid flow and mass conservation, we
perform experiments with and without the enforced positive porosity. Hereafter, these experiments are referred to as 'capped'
and 'uncapped' experiments, respectively.


For the Stokes equations (Eqs. 1 and 2), we use the Creeping Flow (CF) module in COMSOL with the quadratic and linear
elements for the velocity and pressure, respectively. For the heat equation (Eq. 3), we use the Heat Transfer in Fluids (HT)
module with the quadratic and continuous Galerkin finite elements. The standard stabilization methods of the streamline and
crosswind diffusions are used for both the CF and HT modules. For solving the time-dependent equations (Eqs. 1–3), we use

the generalized-alpha method with the direct fully-coupled PARDISO solver.

Eq. 5 is solved using the Transport of Diluted Species (TDS) module with the stabilization method of the streamline diffusion.
Eq. 6 is solved using the TDS module for benchmarking experiment against a semi-analytical solution for 1- and 2-D solitary
waves (Section 3) and the generalized Coefficient Form of PDE (CFPDE) module for all the other experiments (Section 4)

because the CFPDE module is more flexible with the boundary conditions that can be applied (e.g., Weak Contribution option).
We applied the CFPDE module for solving Eq. 6 to test its consistency with the TDS module, and the porosity differences
were smaller than $10^{-9}$ (models not shown). We use the quadratic and continuous Galerkin finite elements for spatial
discretization of Eqs. 5 and 6 and the generalized-alpha method with the direct segregated PARDISO solver to solve the
equations. The Jacobian is updated every time step.


## 3. Benchmarking implementation against a semi-analytical solution for solitary waves

### 3.1 Model setup

Simpson and Spiegelman (2011) derived a semi-analytical solution for a solitary wave which travels in the direction opposite
to the gravity (upward) at a fixed speed ($c$) without changing its shape. The solitary wave is kept stationary under the enforced

downward solid velocity of $c$. We first benchmark our implementation of the equations against the semi-analytical solution for





1- and 2-D solitary waves. This benchmark allows us to verify a successful implementation of the equations in the steady-state limit before conducting time-evolving model experiments for which no analytical solution exists.

The governing equations for the solitary waves are as in Eqs. 5 and 6 without the dynamic pressure and mass transfer terms.

A structured mesh consisting of square elements is used to discretize the model domain. A pseudo 1-D model domain has a (non-dimensionalized) unit height and a width equals to two square element sizes, and the 2-D model domain is a square with a nondimensionalized distance of 1 in both dimensions. To solve the compaction pressure accurately, three or more nodes per compaction length should be used (Dohmen and Schmeling, 2021). We use the element size that is a quarter of the compaction length (i.e., $\delta_0/4$) unless otherwise stated. This is equivalent to a nondimensionalized length of 1/256, which provides 9 nodes

per compaction length and 513 nodes over the domain height. As did in Simpson and Spiegelman (2011), we use $v_{s_0} = v_{l_0}$ and $\frac{h_0^2}{\delta_0^2} = 4096$ in both equations. Compaction pressure of zero and porosity of 1 are prescribed to the top boundary. Since the value of initial porosity field for this problem is large enough ($\geq 1$), we set $\phi_\epsilon$ as 0 (i.e., no regularization). A 'free-flux' boundary condition and zero gradient of the compaction pressure are prescribed to the other boundaries.

The semi-analytic solution for a solitary wave defined for a choice of the triplet ($c$, $n$, $m$) is imposed as an initial porosity field (e.g., Figure 1b); $c$, $n$, and $m$ represent the speed of the solitary wave, permeability exponent, and bulk viscosity exponent, respectively. We calculate the initial solution using the python codes provided in the cookbook (TerraFERMA Cookbook; https://doi.org/10.6084/m9.figshare.1466786.v4) of the open-source finite element code TerraFERMA (Wilson et al., 2017). For accuracy, the solution is calculated at the evenly distributed 513 nodes over the domain height, and the peak porosity is

placed at the center of the model domain. Thus, all the nodes of the 1-D model have the exact values of the solution. For the nodes of the 2-D model domain, the solution is calculated at the nodes along the horizontal and vertical lines that pass through the center of the model domain, and the initial values on the other nodes are approximated by using a cubic spline interpolation. The solutions described on the model domain show hump- and cone-like shapes for 1- and 2-dimensions, respectively. Note that the initial compaction pressure is zero across the entire domain. Finally, a constant time step is set using a Courant number

of 0.5 for the solid velocity, which in this case is identical to the velocity of the solitary wave. We set the final time of the model to 0.5.

### 3.2. Benchmarking results

To quantify the growing error of the solitary wave with time, we calculate the phase shift and phase error of the wave relative

to the semi-analytical solution along the vertical line that pass through the center of the model domain (e.g., Simpson and Spiegelman, 2011). The phase shift is estimated by tracking the location of the peak porosity value of the solitary wave relative to the central node (at a distance of 0.5) by fitting a second-order polynomial to the values at the central node and the nodes





above and below it. The calculation of the phase error consists of two-fold: first, the calculated porosity values at the nodes are interpolated using the piecewise cubic spline to obtain the wave form; then, the wave form is migrated back by the phase shift and the phase error is calculated over the nodes as follows:

$$\sqrt{\sum_{k=1}^{l} \left( \frac{\emptyset_{calc,k} - \emptyset_{anal,k}}{\emptyset_{anal,k}} \right)^2} \tag{14}$$

where $\emptyset_{calc,k}$ is the value of the migrated wave form at the $k$th node, $\emptyset_{anal,k}$ is the corresponding value of the semi-analytical solution at the $k$th node, and $l$ is the total node number.

### 3.2.1. Effect of a choice of the triplet (c, n, m)

We benchmark the model solutions using three solitary wave solutions for different choices of the triplet ($c$, $n$, $m$): ($4$, $2$, $1$), ($5$, $3$, $1$), and ($7$, $3$, $1$). All the experiments show that the solitary wave undergoes a sudden downward migration after the first time step, yielding a negative phase shift in the order of ~$-10^{-7}$ (Figure 2a) owing to the initial zero compaction pressure distribution. Then, it slowly migrates upwards with time. The experiments with a choice of the triplet ($7$, $3$, $1$) show the largest phase shifts: ~$4.4 \times 10^{-7}$ and ~$6.2 \times 10^{-7}$ for the 1- and 2-D experiments at a model time of 0.5, respectively; the phase shifts of the other experiments with choices of the triplets ($4$, $2$, $1$) and ($5$, $3$, $1$) are smaller than $10^{-6}$ after a model time of 0.5. As observed in the phase shift, a sudden large phase error occurs after the first time step (an order of $10^{-6}$ or smaller) and linearly increases with time (Figure 2b). The linear increase in the phase error is likely due to numerical diffusion of the porosity field which tends to smooth out the solitary wave.

### 3.2.2. Effect of the element size

We evaluate the effect of the element size on the solitary waves with a choice of the triplet ($5$, $3$, $1$). In addition to the reference experiment with the element size of $\delta_0/4$ above, we consider element sizes of $\delta_0$, $\delta_0/2$ and $\delta_0/8$ (nondimensionalized lengths of 1/64, 1/128, and 1/512, respectively). All other model parameters are the same as in the reference experiment.

The 1-D experiment using the element size of $\delta_0$ and both the 1- and 2-D experiments using the element size of $\delta_0/8$ show large phase shifts growing with time (results not shown). This is because in these experiments a solitary wave originating from the top boundary passes the model domain toward the bottom boundary, resulting in a substantial change in the shape of the existing solitary wave and forces the wave to migrate upward faster than in the other experiments. To minimize this "passing-solitary wave" effect in these specific models, we used an initial condition field, calculated at the evenly distributed 1025 nodes





over the model height instead of 513 nodes. Using such a refined initial condition, the passing-solitary wave does not occur in either 1- or 2-D experiments except in the 1-D experiment using the coarsest element size of $\delta_0$.

Overall, with the refined initial condition, the absolute net phase shift is relatively small ($\leq 10^{-6}$) for all the element sizes that were tested except for the element size of $\delta_0$, which show a much larger net phase shift (~$10^{-5}$), for the reason detailed above

(Figure 2c). As in the previous experiments, the phase error slowly grows with time likely due to numerical diffusion of porosity field (Figure 2d). Moreover, a decrease in mesh resolution leads to an increase in the error. These experiments confirm that the higher mesh resolution we use, the smaller phase shift and phase error we observe.

## 4. Effect of porosity cap in 2-D time-evolving problems

Although the benchmarking models above verify a successful implementation of the compaction equations, the effects of the capped porosity on the liquid transport and mass conservation should be quantitatively evaluated. In the evaluation, we start with the simplest case where the solid does not flow (stagnant solid) and then apply three solid-flow patterns that are applicable to Earth's mantle: Couette flow, convective flow, and subduction corner flow. Although no analytical solution exists for the modeling schemes, the relatively simple flow patterns that are applied in the models allow reasonable quantification of the

sensitivity of liquid transport and mass conservation to the use of a capped porosity.

To monitor the accuracy of our computations through time, we evaluate the mass balance of liquid (e.g., Lee et al., 2021). Since we assume a constant liquid density, this is equivalent to evaluating the volume balance of liquid. The latter evaluation is two-fold. First, we evaluate the accumulated volume of liquid ($\Phi_{acc}$) over the model domain by removing the initial total

volume of liquid from the calculated total volume of liquid at a given time $t$. Second, we evaluate the net volume flux of liquid ($\Phi_{flux}$) through the model liquid boundaries over a given time $t$. Both evaluations are conducted by assuming a unit thickness of the model domain (i.e., normal to the model domain) (e.g., Lee et al., 2021), as follows:

$$\Phi_{acc}(t) \;=\; \oiiint \big(\phi(t) - \phi(0)\big)\, dV \qquad (14)$$

$$\Phi_{flux}(t) = \int_0^t \left\{ \oiint \left( \frac{v_{l_0}}{v_{s_0}} \phi(t) \boldsymbol{v}_s + K \boldsymbol{k}_{up} \right) \cdot \boldsymbol{n}\, dS \right\} dt \qquad (15)$$

where $\oiiint$ is the volume integral over the model domain, $\phi(t)$ and $\phi(0)$ are porosity at a certain time $t$ and at the initial model

time $t = 0$, respectively. $\oiint$ is the surface integral over the model boundaries. Note that positive and negative values of $\Phi_{flux}$ in Eq. 15 indicate net volume outflux and influx, respectively. For the time-integration of the net volume flux of liquid, we use the trapezoidal rule with a constant time step.





Theoretically, assuming that the time-integration scheme is accurate, the sum $\Phi_{acc} + \Phi_{flux}$ should be equal to zero. In practice

a numerical error is expected to occur owing to classical finite element method itself and numerical integration of the liquid

volume. Hence, we evaluate this relative volume-balance error as:

$$\Delta = \frac{\Phi_{acc} + \Phi_{flux}}{\Phi_{acc}} \times 100 \ (\%) \tag{16}$$

In the subduction corner flow experiment, a liquid source is introduced within the uppermost slab-layer. Thus, taking the liquid

source into account, the relative volume-balance error is evaluated as

$$\Delta = \frac{\Phi_{acc} + \Phi_{flux} - \Phi_{source}}{\Phi_{acc}} \times 100 \ (\%) \tag{17}$$

where $\Phi_{source}$ is the liquid volume created within the uppermost slab-layer over the model time, which is also calculated using

the trapezoidal rule with a constant time step.

### 4.1 Liquid flow through a stagnant porous solid

#### 4.1.1 Model setup

Firstly, we consider a 2-D time-evolving problem with a prescribed porosity at the bottom boundary of a square domain (height

and width are 50, equivalent to 50 km in dimension) of a stagnant (immobile) porous solid (Figure 3a). We solve Eqs. 5 and 6

using the model parameters described in Table 1 and the parameters that are specified below.

The Dirichlet liquid boundary condition at the bottom boundary is specified using a Gaussian function:

$$f(x) = e^{\left(-\frac{(x-x_c)^2}{2\sigma^2}\right)} \tag{13}$$

where $x_c = 25$ (25 km) is the location of the maximum liquid and $\sigma$ is the standard deviation set equal to 1 (1 km). A zero-

compaction pressure gradient (i.e., a zero liquid flux condition) is prescribed to all the boundaries. The initial porosity field is

set 0 over the entire domain.

A constant time step of 0.02 (2000 yr) is used for satisfying the Courant criterion. The liquid flow is calculated for a model

time of 300 (30 Myr). We use square elements of which size is $\delta_0/4$ (i.e., nondimensionalized length of 1/4) for accuracy of

the solution (Dohmen and Schmeling, 2021).

#### 4.1.2 Result

In both capped and uncapped experiments the solitary waves ascend vertically from the bottom boundary. With time, the waves

tend to become vertically elongated, eventually merging into a channel with periodic highs and lows of porosity (Figure 4a

and b). In the uncapped experiment, negative porosity values down to ~-0.08 occur near the ascending wave in the lower model



domain. The negative porosity values tend to disappear as the wave passes through the upper model domain, and after a model time of ~150, the entire porosity field is positive. With time, the porosity fields in both experiments do not reach steady state but exhibit a periodic behavior, resulting in a periodic integrated volume flux at the top wall boundary ($\Phi_{flux,top}$) (Figure 4c). Overall, the evolution of solitary waves in both experiments is similar.

In the capped experiment, the accumulated volume of liquid ($\Phi_{acc}$) first increases until a model time of 50 and then decreases to a stable value at a model time of ~150 (Figure 4d). The accumulated volume of liquid in the uncapped experiment follows similar trends but is consistently lower than that in the capped experiment after a model time of 50. The capped and uncapped experiments share a similar trend in the net volume flux of liquid ($\Phi_{flux}$) through the model boundaries (Figure 4e). The negative sign of net volume flux indicates that the amount of liquid which has entered through the model boundaries over time is larger than what has left through the boundaries, which is thus consistent with the evolution of $\Phi_{acc}$. Here too, the net volume flux stabilizes after a model time of ~150, indicating that the integrated volume influx of liquid is balanced by the integrated volume outflux of liquid through the model boundaries.

The evolution of the relative volume-balance error (Figure 4f) illustrates the influence of the enforced positive porosity in the capped experiment. The enforcement results in an overestimation of the net accumulated volume of liquid until a model time of ~150, yielding an increase in the volume, relative to that in the uncapped experiment (Figure 4d). The negative porosity that is allowed in the uncapped experiment leads to the smaller accumulated volume of liquid and a more accurate volume balance. By a model time of 300, the relative errors of the capped and uncapped experiments are 1.71% and of the order of $10^{-2}$ %, respectively.

## 4.2 Liquid flow through a Couette flow of a porous solid

### 4.2.1 Model setup

Here, we consider the evolution of a liquid flow through a Couette flow of a porous solid from left to right (maximum solid velocity: 3, equals to 3 cm/yr) using the same model domain, boundary conditions, and methods shown in Section 4.1.1 (Figure 3b). No porosity is brought in by the Couette flow across the left boundary.

### 4.2.2 Result

Due to the Couette flow, the solitary waves originating from the bottom boundary are detoured right-wards in both capped and uncapped experiments (Figure 5a and b). With time, the solitary waves tend to form channels displaying periodic highs and lows of porosity, forming very similar peak volume outfluxes of liquid at the right boundary (Figure 5c). However, the



uncapped experiment additionally develops two negative porosity channels under the positive porosity channel, yielding the negative volume outfluxes of liquid at the right boundary (at y-coordinates from 29.62 to 45.52, and from 47.46 to 48.85).


The time-evolution of the integrated volume outfluxes of liquid at the right boundary ($\Phi_{flux,right}$) show that the fluxes asymptotically but slowly converge (Figure 5d). Additional experiments were performed for a longer model time of 1000 and reached steady state (not shown). The capped experiment shows a larger $\Phi_{acc}$ than that in the uncapped experiment from early stages to the end of the model time (Figure 5e). In particular, the asymptotic value of $\Phi_{acc}$ in the capped experiment remains

approximately 30% higher than that in the uncapped experiment. The net volume fluxes of liquid show a pronounced divergent behavior between the two experiments from a model time of ~25 (Figure 5f). While $\Phi_{flux}$ in the capped experiment increases with time from negative values in the early stage, indicating influx, to positive values in the later stage, indicating outflux, $\Phi_{flux}$ in the uncapped experiment stabilizes to an apparent steady state. This indicates that in the capped experiment, the integrated volume outflux exceeds the integrated volume influx through the model boundaries. On the contrary, the stable

value of $\Phi_{flux}$ in the uncapped experiment indicates that the integrated volume influx of liquid is balanced by the integrated volume outflux of liquid through the model boundaries. Both $\Phi_{acc}$ and $\Phi_{flux}$ in the uncapped experiment are well balanced and the relative volume-balance error remains very small (0.09% at a model time of 300). Similar to the behavior of $\Phi_{flux}$, the relative volume-balance error in the capped experiment largely and continuously increases with model time (reaching 332% at a model time of 300) even after the model starts to reach steady state whereas the error in the uncapped experiment

diminishes with time (Figure 5g).

The enforced positive porosity in the capped experiment leads to a larger net accumulated volume of liquid compared to that in the uncapped experiment in which the accumulated volume is counterbalanced by negative porosity. Thus, $\Phi_{acc}$ is overestimated in the capped experiment relative to the uncapped experiment, and the higher accumulated porosity leads to an

overestimation of the integrated volume outflux through the right boundary as seen in the distribution of volume flux (Figure 5c) and the integrated value over time (Figure 5d). The increase in the outflux is not sufficient to completely offset the increase in the accumulated porosity, and as a consequence, the accumulated porosity, the net volume (out)flux of liquid through the model boundaries, and the relative volume-balance error continuously increase (Figure 5e and f).

**4.3. Liquid flow through convecting porous solid**

**4.3.1 Model setup**

Here, we consider the evolution of the liquid flow through convecting porous solid using the same model domain, boundary conditions, and methods shown in the section 4.1.1 (Figure 3c). We apply free-slip to all four boundaries for solving the solid



flow. For solving the heat equation, the top and bottom temperatures are fixed as 0 and 1 (0 and 1000 °C), respectively, and

the left and right walls are prescribed with a zero-heat flux boundary condition. The quasi-steady state convecting flow is first

calculated by solving Eqs. 1–3 for a model time of 500 (50 Myr). Then, the liquid flow is calculated for a model time of 300

(30 Myr). No porosity influx by solid advection is allowed across all four boundaries while porosity outflux is allowed (i.e., a

free-outflux and zero-influx boundary condition).

### 4.3.2 Results

Due to the clockwise solid convection, the solitary waves ascending from the bottom boundary are detoured left-wards and

right-wards in the lower and upper model domains, respectively, in both capped and uncapped experiments (Figure 6a and b).

Most of the ascending liquid leaves the domain through the top boundary but a fraction of it continues to be entrained in the

convecting solid and remains within the domain. The entrained liquid then merges with newly ascending liquid from the bottom

boundary. With time, the waves tend to form a quasi-steady-state channel in the capped experiment, which leads to a quasi-

steady-state integrated volume outflux of liquid (but slowly increasing) through the top boundary ($\Phi_{flux,top}$) (Figure 6c). On

the contrary, the porosity field in the uncapped experiment does not converge to steady state by the end of the model time.

This also leads to an unsteady integrated volume outflux of liquid through the top boundary. Negative porosity values occur

near the liquid pathway by the end of the model time, decreasing the integrated volume outflux of liquid at the top boundary.

In both capped and uncapped experiments, the evolution of $\Phi_{acc}$ can be divided into three stages (Figure 6d): a first transient

stage (0 to ~40) corresponding to times when the first solitary wave ascends through the model domain and reaches the top

boundary. A second transient stage (~40 to ~120) corresponds to times when a fraction of liquid of the first solitary wave is

advected downwards by the convecting cell and merges with solitary waves that have entered the domain and ascended from

the bottom boundary. The second stage ends when these merged solitary waves reach the top domain. A third stage (~120 to

the end) corresponds to the period in which the entrained liquid continues to merge with the solitary waves that ascend from

the bottom boundary; the rotating porosity channel reaches semi-steady-state. The capped experiment shows an increasing

$\Phi_{acc}$ at all of these stages even in the third stage, whereas the uncapped experiment shows a stable value during the third stage.

On the contrary, the net volume fluxes of liquid ($\Phi_{flux}$) are equally stable (very slowly decreasing) in both experiments after

a model time of ~150 in the third stage (Figure 6e). The relative volume-balance error ($\Delta$) in the capped experiment displays

a net increase with time and reaches 17.3% at a model time of 300. However, in the uncapped experiment, it slowly decreases

with time and reaches -2.8% at the same time (Figure 6f).

As in sections 4.1 and 4.2, the enforced positive porosity in the capped experiment results in an overestimation of $\Phi_{acc}$. The

overestimation is relatively small until a model time of ~30 but becomes significantly larger when the first porosity wave



reaches the top boundary from a model time of ~30 to ~40. Between model times of ~30 and ~40 we also observe a slightly

higher $\Phi_{flux}$ in the capped experiment which indicates that the outflux is slightly larger than that in the uncapped experiment

owing to the enforced positive porosity. The relative volume-balance error also undergoes a fast increase between model times

of ~30 and ~40 in the capped experiment, which is not observed in the uncapped experiment. During the second stage, the

impact of the enforced positive porosity on the $\Phi_{acc}$ is relatively small as illustrated by the similar evolution of the $\Phi_{acc}$

between the two experiments before a model time of ~120. Thus, both experiments show stable relative volume-balance errors

though a slight decrease of the relative volume-balance error with time is observed in the uncapped experiment. During the

third stage, however, large divergences of $\Phi_{acc}$ and relative volume-balance error occur between both experiments. The

enforced positive porosity mostly impacts $\Phi_{acc}$ which undergoes a substantial increase with time in the capped experiment.

That is because the overestimated $\Phi_{acc}$ progressively impacts newly ascending liquid volumes. On the contrary, although

$\Phi_{flux}$ remains always slightly larger in the capped experiment, the difference in $\Phi_{flux}$ between both experiments remains

stable throughout the third stage. This shows that the continuous increase in relative volume-balance error recorded in the

capped experiment is solely due to the overestimation of $\Phi_{acc}$. As observed in the previous models (sections 4.1 and 4.2), the

volume balance of the uncapped experiment is better kept.

### 4.4 Liquid flow through a subduction corner flow

**4.4.1 Model setup**

Lastly, we consider the evolution of the liquid flow through a subduction corner flow in which the solid-viscous flow in the

corner wedge is kinematically driven by the subducting slab. The height and width of the model are 50 and 52.8 (equal to 50

and 52.8 km), respectively (Figure 3d). The subducting slab has a dip of 45 degrees and a subduction rate of 5 (5 cm/yr).


For solving the solid flow, free-slip and open boundary conditions are prescribed to the top and right boundaries of the mantle

wedge, respectively (Figure 3d). To reach a steady-state corner flow in the mantle wedge, Eqs. 1 and 2 without the buoyancy

term in Eq. 2 are solved for a model time of 500 (50 Myr).

For solving the liquid flow, a free-outflux and zero-influx boundary condition is prescribed to all the boundaries except for the

base of the uppermost slab layer which is prescribed with a zero porosity (Figure 3e). A zero gradient of compaction pressure

is prescribed to all the boundaries. A Gaussian source term (same function as in Eq. 13) for $\Gamma$ in Eqs. 5 and 6 is applied within

the uppermost subducting slab over a thickness of 2 (2 km). The source term is a simplified proxy for a dehydration reaction

that would produce some liquid within the top layer. To discretize the slab and wedge geometry, we use an unstructured mesh

consisting of triangular elements that are slightly smaller than a quarter of the compaction length. The liquid flow is calculated

by solving Eqs. 5 and 6 for a model time of 300 (30 Myr) with a constant time step of 0.004 (400 yr).



**4.4.2 Results**

The general trends of the liquid flow in the wedge are similar in both capped and uncapped experiments, both of which tend

towards a stable dynamics after a model time of 50. Due to the downdip solid flow, the solitary waves originating from the top

layer tend to be slightly detoured right-wards as they ascend through the bottom half of the wedge. In the top half of the wedge,

the inward (leftward) corner flow exerts leftward advection of the ascending waves before they reach the top boundary (Figure

7a and b). Although most of the liquid passes upward through the wedge, a fraction of it is entrained by the corner flow and

leaves the model domain across the right boundary.


Both capped and uncapped experiments yield periodic high and low volume fluxes at the top and right boundaries ($\Phi_{flux,top}$

and $\Phi_{flux,right}$) which correspond to newly arriving solitary waves at the top after ascent and to downdip advection of its

fraction, respectively (Figure 7c and d). The amplitudes and periods of the volume fluxes at the top boundary remain very

similar in both experiments. At the right boundary, the periods of the volume fluxes are also very similar, but the amplitude at

the right boundary is larger in the capped experiment.

Compared to that in the capped experiment, the uncapped experiment shows lower $\Phi_{acc}$ due to the negative porosity field in

the model domain. Similarly, $\Phi_{flux}$ is also lower due to lower outflux at the right boundary (Figure 7e and f). As a result, the

difference in $\Phi_{flux}$ between both experiments increases with time. The capped experiment yields a relative volume-balance

error of 337.17% at a model time of 300 whereas it is only -1.85 % in the uncapped experiment. (Figure 7g). Thus, a better

balance between $\Phi_{acc}$ and $\Phi_{flux}$ is maintained when negative porosity is allowed in the uncapped experiment.

**4.5 Effect of element size on relative error in time-dependent problems**

To check the sensitivity of our results to the choices of element sizes, we run additional experiments with the square element

sizes of $\delta_0/2$ and $\delta_0/8$ (nondimensionalized length of 1/2 and 1/8, respectively), which provide 5 and 17 nodes per compaction

length, respectively, for the experiments of stagnant solid, Couette flow and convective flow. For the experiment of the

subduction corner flow, the triangular elements of which sizes are slightly smaller than $\delta_0/2$ and $\delta_0/8$ are additionally

considered. We calculate the relative volume-balance error at a model time of 300 for all the experiments.

The experiments of stagnant porous solid show a significant decrease in the relative volume-balance error with increasing

mesh resolution in the uncapped experiments (Figure 8a). The error in the capped experiments also decreases with increasing

mesh resolution, but the decrease is much less than an order of magnitude of error and is negligibly small in comparison to the



decrease in the error for the uncapped experiments. Except for the coarsest resolution (element size of $\delta_0/2$), the uncapped experiments warrant better volume balances at all mesh resolutions.


All the other experiments (Couette flow, convecting flow, and subduction corner flow) show the same systematic change in the relative volume-balance error with mesh resolution. In all cases, the allowance of negative porosity in the uncapped experiment always warrants a better liquid volume conservation (i.e., a lower absolute value of the error by several orders of magnitude). In other words, the increase in mesh resolution in the capped experiments does not reduce the large error resulting

from the overestimated porosity in the model domain.

## 5. Discussion and conclusion

In this study, we first conducted a series of benchmarking experiments against a semi-analytical solution for solitary waves (Simpson and Spiegelman, 2011). Although the specifics of the numerical approach used in the study (non-linear solvers, finite

element order, continuous Galerkin finite elements, etc.) differ from those of the previous studies (e.g., Simpson and Spiegelman, 2011; Wilson et al., 2017; Wang et al., 2019), we obtain a relatively accurate solution when the element size is $\delta_0/2$ or smaller (i.e., 5 or more nodes per compaction length). The impact of element size on the solution accuracy is consistent with previous studies (Dohmen and Schmeling, 2021), and as expected, increasing mesh resolution (decreasing element size) significantly improves the solution accuracy partly owing to reduced numerical diffusion of porosity.


Next, we performed time-evolving experiments. Our capped experiments show the best accuracy in mass conservation in models where the background mantle is stationary (Figures 4a and 8a), after the early stages of the models. In other experiments where the solid flows (Couette flow, convective flow, and subduction corner flow), the error in mass balance is not negligible owing to the porosity overestimation induced by the capped porosity and subsequent overestimation of the mass flux across

the model boundaries. Because the overestimation is intrinsic, increasing mesh resolution does not significantly reduce the error. Thus, the estimated volume of liquid in the model domain and volume flux of liquid through the model boundaries (e.g., amount of magma in the mantle and magma migration through the overlying lithosphere) should be carefully interpreted when the solid phase is deforming. We thus emphasize that for future applications, caution should be taken when using the melt and flux estimations. However, the general porosity behavior and porosity peaks (largest positive liquid fractions) in the model

domain and boundaries are quite similar in both the capped and uncapped experiments. Hence, the use of a capped porosity in the compaction equations is reasonable for estimating the liquid pathways and first-order distribution of fluids and melts in the mantle beneath arcs and ridges (e.g., Wilson et al., 2014; Sim et al., 2020; Cerpa et al., 2017). Models of liquid transport in the mantle with capped porosity field may be compared to geophysical imaging (e.g., magneto-tellurics and seismic



tomography) of the asthenosphere which illuminates the first-order distribution of fluids and melts, for example, in the sub-
arc mantle wedge of subduction zones (Mcgary et al., 2014; Cordell et al., 2019; Bie et al., 2022).

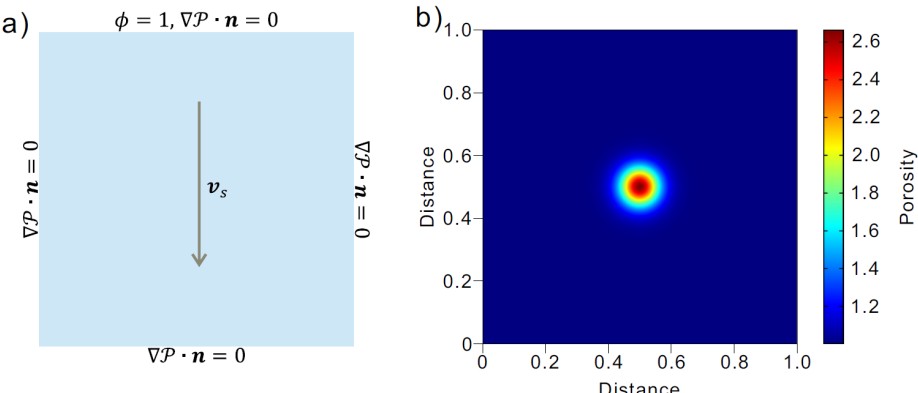

**Figure 1. a) Model boundary conditions used for both the 1- and 2-D experiments. b) Initial porosity from the 2-D**
**experiment with a choice of the triplet (5, 3, 1).**

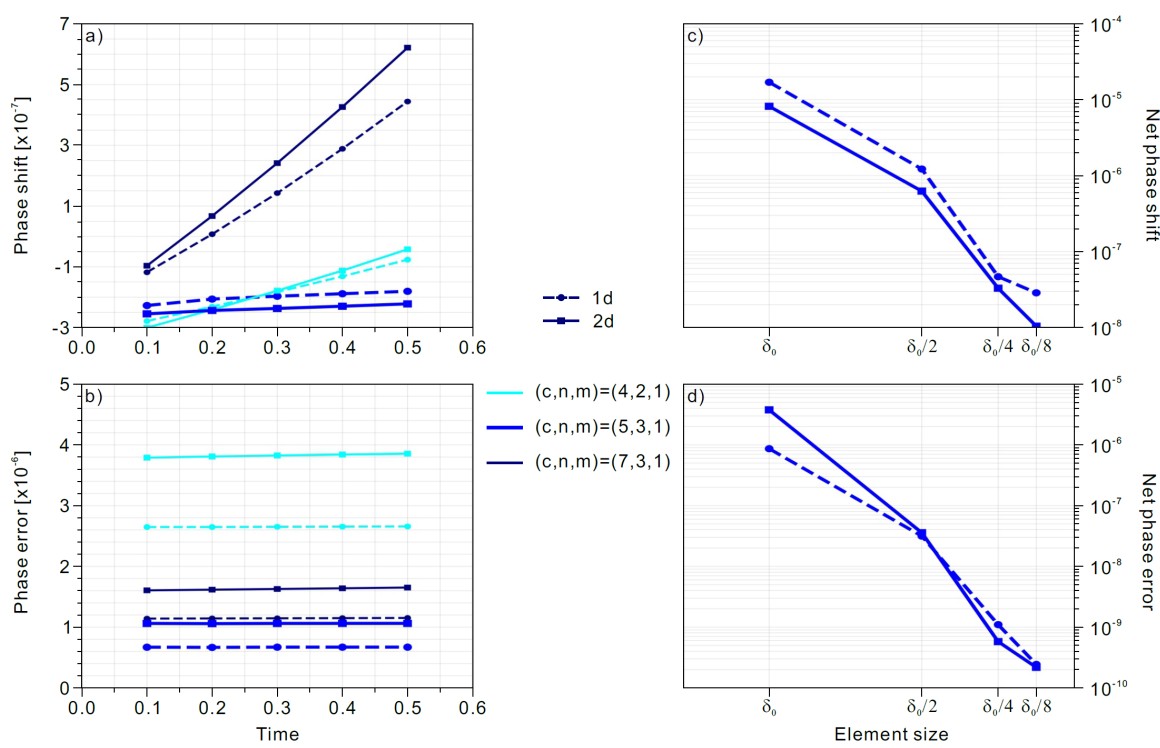



**Figure 2. a and b) Time-evolution of the phase shifts and phase errors for the calculated solitary waves for three choices of the triplets (c, n, m) and the 1- and 2-D experiments. c and d) Absolute net phase shifts and phase errors of the**

**calculated solitary waves with a choice of the triplet (5, 3, 1) for element sizes of $\delta_0$, $\delta_0/2$, $\delta_0/4$, and $\delta_0/8$ and for the 1- and 2-D experiments, both calculated over the model time period 0.4 (from a model time of 0.1 to 0.5).**

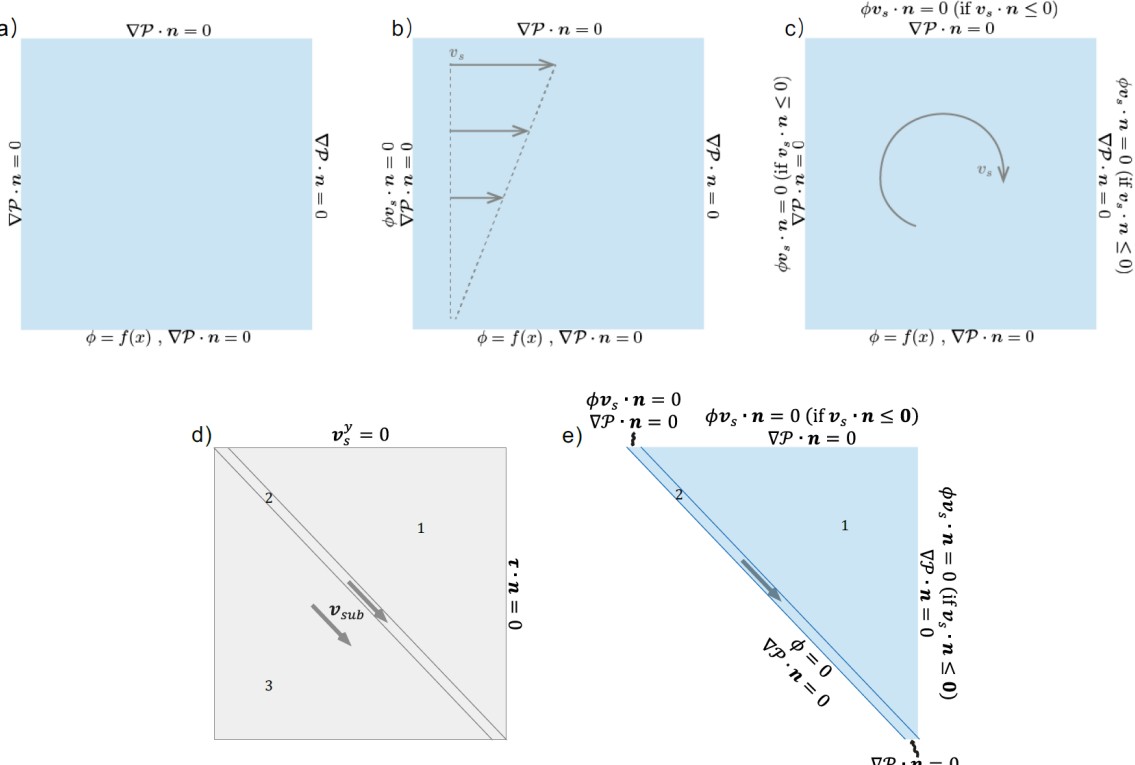

**Figure 3. Model boundary condition used for the 2-D models of two-phase flow with four different solid flow patterns: (b) Couette flow, (c) convective flow, and (d and e) subduction corner flow.**



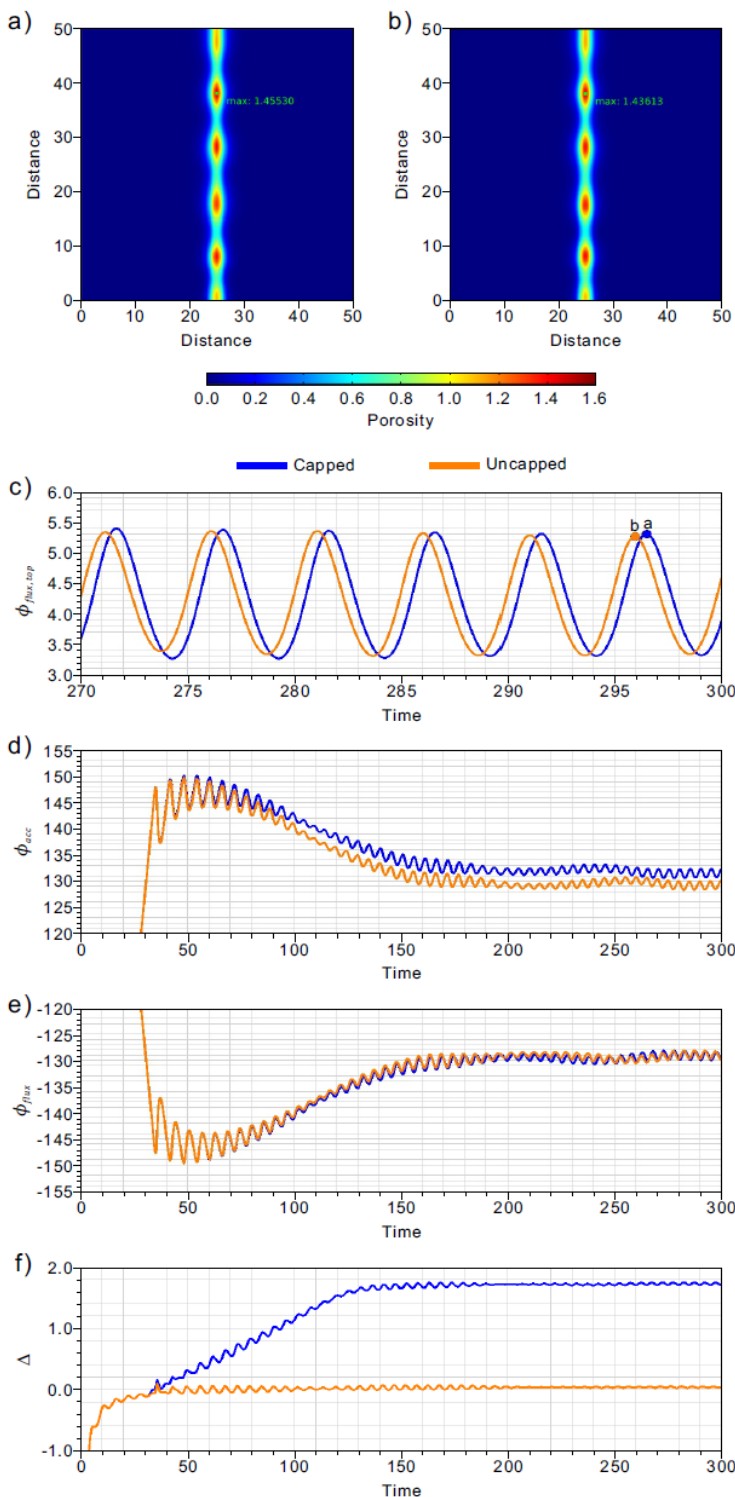



**Figure 4. a and b) Snapshots of the porosity fields from the capped and uncapped experiments at model times of 296.50 and 295.96, respectively. The maximum porosities are depicted. c) Evolution of the integrated volume flux through the top boundary ($\Phi_{flux,top}$) from both experiments from a model time of 270 to a model time of 300. d) The net accumulated volume of liquid ($\Phi_{acc}$) over the model domain from both experiments from a model time of 0 to 300. e) The net volume flux of liquid ($\Phi_{flux}$) through the model liquid boundaries from both experiments from a model time of 0 to 300. f) The relative volume-balance error ($\Delta$) from both experiments from a model time of 0 to 300.**





**Figure 5. a and b) Snapshots of the non-dimensional porosity fields from the capped and uncapped experiments at model times of 298.02 and 298.92, respectively. The maximum and minimum porosities are depicted. c) Volume flux over the right boundary from both the experiments at the model times shown in a and b. d) Evolution of the integrated volume flux through the right boundary ($\Phi_{flux,right}$) from both experiments from a model time of 270 to**



**300. e) The net accumulated volume of liquid ($\Phi_{acc}$) over the model domain from both experiments from both experiments, f) the net volume flux of liquid ($\Phi_{flux}$) through the model liquid boundaries from both experiments, and g) the relative volume-balance error ($\Delta$) from both experiments from a model time of 0 to 300.**



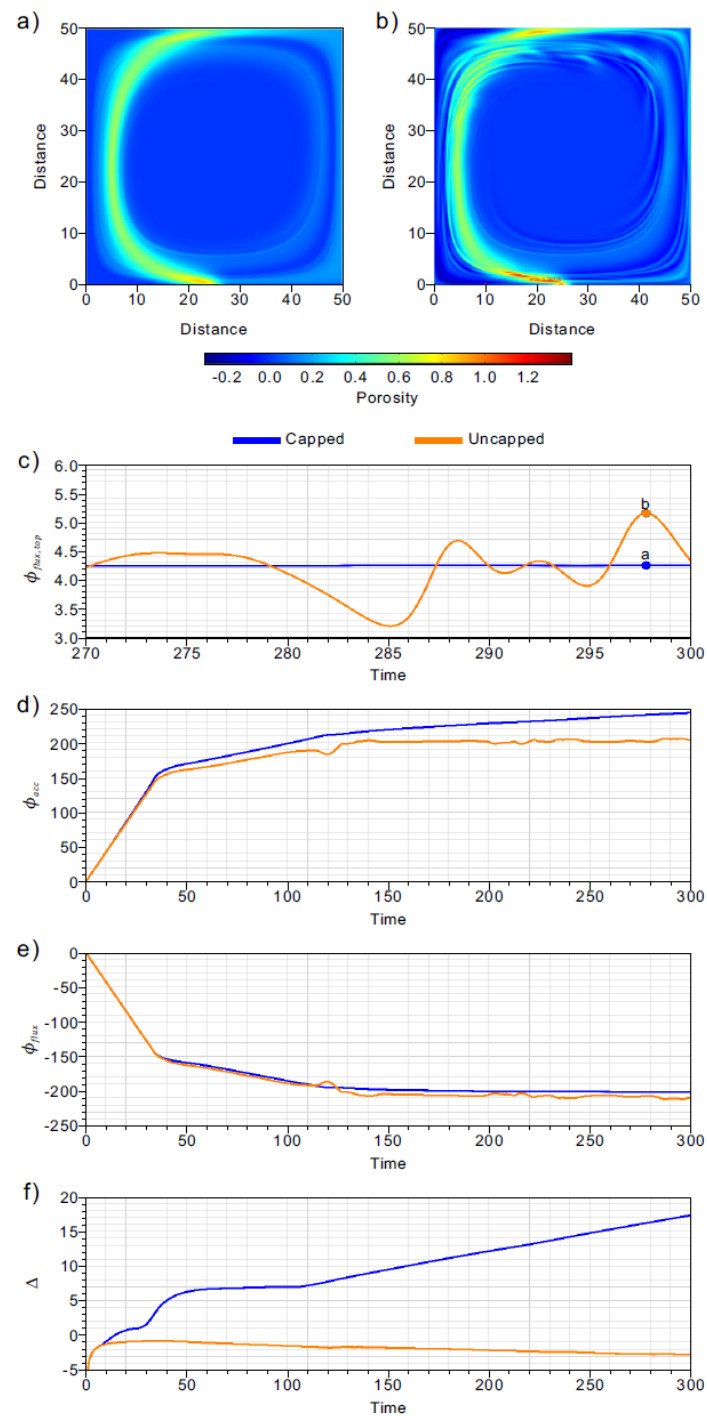

**Figure 6. a and b) Snapshots of the non-dimensional porosity fields from the capped and uncapped experiments at a**
**model time of 297.80. The maximum and minimum porosities are depicted. c) Evolution of the integrated volume flux**





through the top boundary ($\Phi_{flux,top}$) from both experiments from a model time of 270 to 300. d) The accumulated volume of liquid ($\Phi_{acc}$) over the model domain from both experiments, e) the net volume flux of liquid ($\Phi_{flux}$) through the model liquid boundaries from both experiments, and f) the relative volume-balance error ($\Delta$) from both experiments from a model time of 0 to 300.




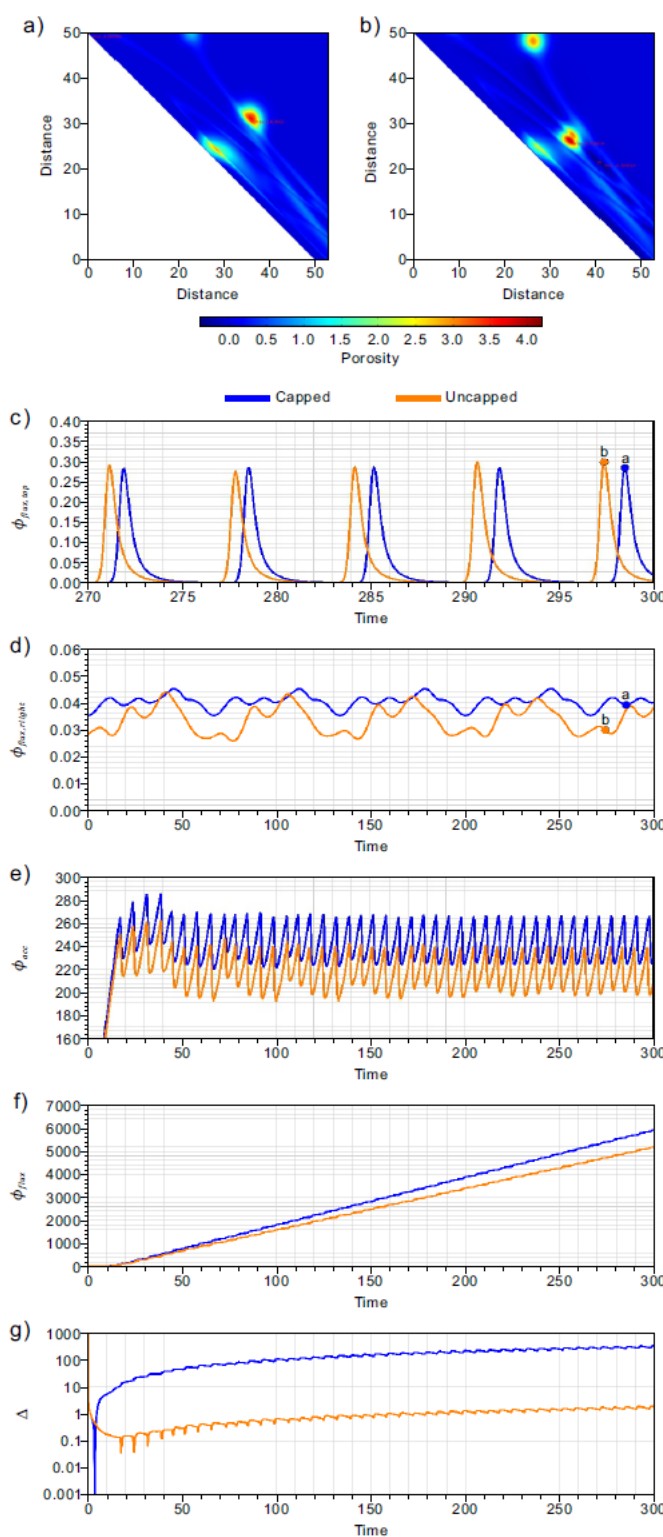



**Figure 7. a and b) Snapshots of the non-dimensional porosity fields from the capped and uncapped experiments at model times of 298.52 and 297.42, respectively. The maximum and minimum porosities are depicted. c and d) Evolution of the integrated volume flux through the top and right boundaries ($\Phi_{flux,top}$ and $\Phi_{flux,right}$) from both experiments**
**from a model time of 270 to 300. e) The accumulated volume of liquid ($\Phi_{acc}$) over the model domain from both experiments, f) the net volume flux of liquid ($\Phi_{flux}$) through the model liquid boundaries from both experiments, and g) the absolute volume-balance relative error ($\Delta$) from both experiments from a model time of 0 to 300.**

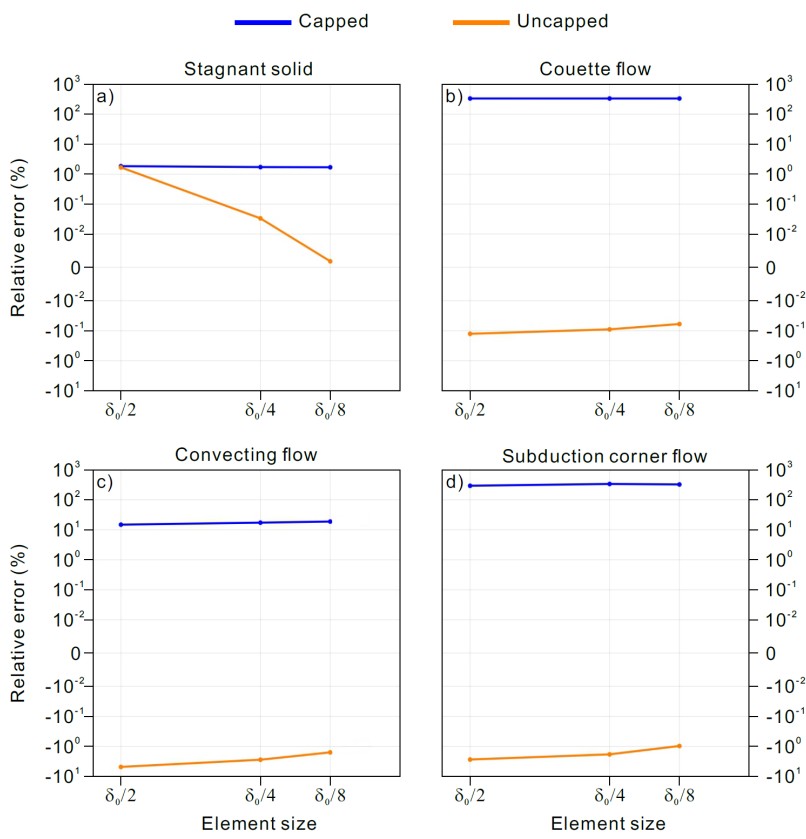

**Figure 8. Relative volume-balance error calculated from the capped and uncapped experiments varying element sizes as $\delta_0/2$ and $\delta_0/8$: a) stagnant solid, b) Couette flow, c) convecting flow, and d) subduction corner flow.**



**Table 1: Model parameters**

| Parameter | Formula | Value |
|---|---|---|
| Reference length $h_0$ (m) | | 1000 |
| Reference solid density $\rho_{s_0}$ (kg m$^{-3}$) | | 3300 |
| Thermal expansivity $\alpha_0$ (K$^{-1}$) | | $2.5 \times 10^{-5}$ |
| Temperature difference $\Delta T$ (K) | | 1000 |
| Gravitational acceleration $g_0$ (m s$^{-2}$) | | 10 |
| Reference solid shear viscosity $\eta_0$ (Pa s) | | $2.1543 \times 10^{19}$ |
| Reference solid velocity $v_{s_0}$ (m s$^{-1}$) | | $3.1710 \times 10^{-10}$ |
| Unit vector in the direction opposite to the gravity $k_{up}$ (.) | | 1 |
| Peclet number $Pe$ | $h_0 v_{s_0}/\kappa_0$ | 0.41857 |
| Thermal diffusivity $\kappa_0$ (m$^2$ s$^{-1}$) | | $7.5758 \times 10^{-7}$ |
| Reference liquid velocity $v_{l_0}$ (m s$^{-1}$) | $K_0 \phi_0^{n-1} \Delta \rho g_0$ | $9.2832 \times 10^{-10}$ |
| Reference liquid mobility $K_0$ (m$^3$ s kg$^{-1}$) | | $1 \times 10^{-8}$ |
| Reference porosity $\phi_0$ (.) | | 0.0021544 |
| Density contrast $\Delta \rho$ (kg m$^{-3}$) | $\rho_{s_0} - \rho_{l_0}$ | 2000 |
| Reference liquid density $\rho_{l_0}$ (kg m$^{-3}$) | | 1300 |
| Reference compaction length $\delta_0$ (m) | $\sqrt{K_0 \phi_0^{n-m} \eta_0}$ | 1000 |
| Permeability exponent $n$ | | 3 |
| Bulk viscosity exponent $m$ | | 1 |
| Background porosity $\phi_b$ (.) | | 0.0021544 |

**Code availability.** The models were run with the commercial finite-element package, COMSOL Multiphysics®
(https://www.COMSOL.com).

**Data availability**. The data used to generate the figures are shared in https://doi.org/10.5281/zenodo.8179527 (Lee et al.,
2023).


**Author contributions**. CL and NC conceived the study. CL and NC designed and ran the models, analysed the results, and
wrote the article. DH revised the python code for the benchmarking experiments and produced figures. IW wrote the article.
All authors discussed the results and their consequences, and contributed to the writing of the final article.



**Competing interests**. The contact author has declared that none of the authors has any competing interests.

**Disclaimer**. Publisher's note: Copernicus Publications remains neutral with regard to jurisdictional claims in published maps and institutional affiliations.

**Acknowledgements**. This study was funded by the National Research Foundation of Korea nos. 2017R1A6A1A07015374 and 2022R1A2C1004592.

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
