# Peer review of "Modeling liquid transport in the Earth's mantle as two-phase flow: Effect of an enforced positive porosity on liquid flow and mass conservation"

_EGUsphere, 2023_

## Referee Comment (RC2)

[referee-annotated manuscript omitted]

---

## Author Response (AR1)

Dear Editor, dear reviewers,

Please find below our responses (in blue) to the reviewer's comments. The line numbers refer to those in the revised document (in red) that we also attach to our response.

Sincerely,

Changyeol Lee (on behalf of all co-authors)

Referee #1

The paper discusses an interesting and important topic related to the issue of negative porosity appearing in numerical simulations.

Some issues that should be addressed follow.

1. Abstract, line 17: "The implementation of these equations requires ..." Perhaps it is best to soften the statement, since there are "positivity preserving" numerical methods used in other fields. Such methods may be adaptable to mantle dynamics.

>A: We have softened the statement by modifying it to "When implementing these equations, it is **common** to use a regularization technique to handle the porosity singularity in dry mantle" (line 17).

2. Line 115: The singular behavior is due to (10) and (11), not just (10).

>A: As the reviewer suggested, we have corrected this and now say that the singular behavior is due to both (line 122). Please note that the equation numbers 4 and 8 were missing, and thus we also corrected the equation numbers.

3. Lines 122-124: I assume that you mean that the computed phi is replaced by max(0,phi). Please define precisely what it means to "cap" the porosity, so there is no confusion.

>A: In the experiments with an enforced positive porosity, i.e., the "capped" porosity experiments, we impose $\phi = \max(0, \phi)$. We modified the text accordingly (lines 131 and 148-149).

4. Line 151: "equals" should be "equal".

>A: We rephrased the text for clarity (lines 162-163).

5. Line 157: This is the first statement that the porosity is greater than 1. It also occurs in the numerical results. Since the porosity can not exceed 1 by definition, perhaps some explanation is needed here.

A: That is because the porosity is scaled to some reference porosity $\phi_0$ (the dimensional porosity is $\phi_{dimensional} = \phi_0 \phi \leq 1$). For clarity, we rephrased the sentence (lines 167-168). Moreover, we added a sentence "In what follows, until stated otherwise, we always describe non-dimensional values (porosity, compaction pressure, time, etc.)." to avoid further confusions (lines 183-184).

6. Example 4.1: The explanation of results seems to miss the obvious. Figure (e) implies that the boundary fluxes agree, so (d) is due solely to capping. The amount of capping is significant.

A: We have placed such an explanation when describing Fig. 4f. The relative volume-balance error increases until a time of ~150 and then remains stable, reaching a relatively small value of 1.71% at a model time of 300 (lines 294-299). In response to the reviewer's comment, we have added one sentence to explain that the difference in the accumulated porosity is solely due to capping (lines 286-288).

7. Examples 4.2-4: It is difficult to compare the two numerical results in these problems. Figure (g) implies a large capping effect, while (b) shows a large error in the uncapped result (there is a lot of negative porosity). Thus it is difficult to determine the correct behavior. Can you run a refined example to determine the correct behavior of the systems?

A: As suggested, we additionally considered the experiments with a finer resolution using an element size of $\delta_0$/16 (lines 428-430), which is now discussed in Section 4.5. We found that the overall porosity evolution is similar regardless of the element size, but with increasing resolution, the capped and uncapped experiments converge towards different solutions. That is because the capped experiments raise an intrinsic error resulting from the enforced positive porosity in the model domain as explained throughout Sections 4.2 to 4.4. For clarity, we thoroughly revised Section 4.5 (lines 434-449).

8. Line 440: The phrase "increasing mesh resolution does not significantly reduce the error" seems incorrect, and the statement is unsubstantiated. Please run some examples to either show whether this is the case.

A: As shown in the element size tests (Section 4.5), the intrinsic error due to the enforced positive porosity cannot be removed by increasing the mesh resolution. For clarity, we rephrased the relevant sentence (lines 463-464)

9. The conclusion that it is reasonable to cap the porosity (appearing several places, such as the abstract and conclusions) seems a bit strong. Only the overall system behavior agrees between the two numerical methods in the tested scenarios. But the behavior also differs substantially in important ways.

A: We have modified the abstract (lines 28-31) and conclusion (lines 467-470) to take into account the reviewer's suggestion. We argue that the positive porosity and peaks (largest positive liquid fractions) are similar in both the capped and uncapped experiments, so the use of capping appears reasonable to assess the main fluid pathways and first-order distribution of fluids and melts in the mantle.

Referee #2 Samuel Butler

A review of "Modeling liquid transport in the Earth's mantle as two-phase flow: Effect of an enforced positive porosity on liquid flow and mass conservation"

In this paper, the authors solve the compaction equations in the small porosity limit in a number of scenarios, including solitary waves, in a fixed solid background, in a Couette flow solid background, in a convecting solid background and in a corner flow. For the solitary waves, the authors compare their solutions to analytical solutions while for the other cases, the authors monitor the quality of their solutions by considering the global conservation of liquid. The authors compare solutions in which the porosity is "capped" or forced to remain between 0 and 1 and "uncapped" in which case it sometimes becomes negative. They show that conservation of liquid is generally better obeyed for the uncapped solutions. However, they conclude that the general behaviour of the capped and uncapped solutions is similar and that capped solutions are generally reasonable.

The paper explores an interesting range of models and is generally well written. I have made a number of comments on the annotated manuscript that I would like for the authors to act on.

I have the following additional comments:

1. The main purpose of the paper is to compare the behaviour of "capped" and "uncapped" solutions. However, the authors do not describe in the paper how they "cap" the solutions. If the porosity solution becomes negative during time stepping, do they simply set it to 0 in the model? Also, how is the capping implemented in Comsol? Both of these questions need to be addressed.

A: To enforce positive porosity in the capped experiments, we impose $\phi = \max(0, \phi)$ using the Lower Limit option in the solver (lines 131 and 148-149)

2. Additionally, I would think that the error would depend on how the capping is carried out. Butler (2017) introduced an analytical transformation that caused the porosity to always fall between 0 and 1. I think that authors should compare this type of "capping" and perhaps others in terms of the liquid volume conservation errors.

A: We now acknowledge the approaches that are used in Wilson et al. (2014), Butler (2017), and Sim et al. (2020) in the line 124. However, we've chosen to focus on estimating the liquid volume error that are associated with the regularization technique that has been used several times in previous studies (Wilson et al., 2014; Cerpa et al., 2017, 2018) (lines 126-127). One good thing about this technique is that it allows to

control the amount of regularization according to a desired minimum compaction length which can then be chosen based on the model resolution. Evaluating the errors associated with alternative techniques, although useful, is out of the scope of the present manuscript.

3. The authors only consider conservation of liquid volume as an error measure. However, they could also consider the global balances represented by the other governing equations of their system. For instance, integrating equation 2 over the domain gives a mechanical energy balance equation i.e. the work done on the system by forces exerted on the boundaries is balanced by the work done by gravity. Including these additional balances would significantly increase the level of interest in this paper.

A: Our goal was to evaluate the implementation of the equations for liquid flow under the assumption of the small-porosity limit, which leads to decoupling of the solid from liquid flow (see description in the lines 90-92). Therefore, we carried out the analyses of the accuracy of the solutions to these compaction equations through an evaluation of the mass conservation of liquid. We now highlight this in the lines 129-131.

That said, the accuracy of the solution of the solid-state mantle flow (Stokes equation) has been evaluated in depth in previous studies. Thus, we now refer to previous published works (line 77). For instance, Lee (2013) shows that the work against gravity is well balanced by viscous dissipation even in the compressible fluid. The convection models shown in Trim et al. (2021) also show good agreements with the reported Nusselt numbers and mean temperatures.

4. Line 52: delete "a" from a continuum mechanics

A: We deleted (line 55).

5. Line 72-76: In a series of papers, Butler investigated compaction-driven porosity segregation using Comsol.

A: Thank you for the reference. We added it to the lines 79-80.

6. Line 110: is mobility different from permeability?

A: The liquid mobility is defined as a permeability divided by liquid viscosity (line 117). It has units of $m^3$ s $kg^{-1}$ (see Table 1)

7. Line 112-114: I find this logic a bit strange. Do these really follow from above? I would give these equations before equations 7 and 9. Also, there doesn't seem to be an equation 8.

A: Eq. (8)-(10) are consequence of our non-dimensionalization choices with non-dimensional terms partly given in Eqs. (6) and (7). We slightly re-organized this section and rephrased the sentence to avoid confusion (line 120).

8. Line 123: Butler (2017) introduced a transformation that forced porosity to always be positive.

A: We added Butler (2017) to acknowledge an alternative approach to avoid porosity singularity (lines 123-124)

9. Line 124: Please explain how "capping" is implemented in Comsol.

A: We used the Lower Limit option. It is now said in the lines 148-149.

10. Line 149: Please explain the assumptions made that allow the dynamic pressure and mass transfer term to be neglected.

A: In Section 3.1, we present the modeling results for the solitary waves that are described by Simpson and Spiegelman (2011), in which the mass transfer between solid and fluid phases, as well as the gradients of the dynamic pressure, are neglected. These are assumptions made by Simpson and Spiegelman in deriving the analytical solution, to which we are comparing our results. For clarity, we rephrased the sentence (line 159).

That said, for the remainder models (time-evolving), we follow previous studies (e.g., Wilson et al., 2014; Cerpa et al., 2017) that also neglect the gradients of the dynamic pressure in the compaction equation (Eq. 5). As discussed in those studies, the magnitude of dynamic-pressure gradients should be negligible compared to those of the compaction pressure in most areas of interest (e.g., in core of a convection-cell, in most of the mantle wedge corner, etc.). We have added a few lines for commenting this (lines 106-108).

The rate of mass transfer ($\Gamma$) is only considered in the subduction corner flow experiment, to implement the dehydration reactions. For clarity, we rephrased the relevant sentence (lines 263-264), which clarifies a choice of mass transfer term in Sections 4.2, 4.3, and 4.4 (lines 303-305, 340-341, and 400-401, respectively).

11. Line 155-156: This information should be given before the description of the discretization. Also, it would be worth mentioning to the reader that since the system is set to be large compared with the compaction length, that the effects of compaction are likely to be significant.

A: We agree with the reviewer. Thus, we have moved the sentence such that it appears before the description of discretization with the additional description of the domain size compared to the compaction length (lines 159-161)

12. Line 157: Please give the mathematical form of the "free-flux" boundary condition

A: We impose the unit porosity ($\phi = 1$) and the zero gradient of the porosity ($\nabla\phi \cdot n = 0$) at the top boundary and the other boundaries, respectively. To avoid confusion, the description of the boundary conditions has been rephrased (lines 168-170) together with the revised Figure 1.

13. Line 211-212: It would be useful to discuss here how long models can be run before the accumulating numerical error becomes a problem. Can the authors scale their problem to an Earth-relevant time scale and compare?

A: The solitary-wave benchmark is not designed for a specific spatiotemporal scale of geological interests (lines 159-160). That said, using the scaling parameters shown in Table 1, we observe that the benchmark model remains relatively accurate up to a modeled time of 0.05 Myr. We added some sentences in the lines 227-229.

14. Line 377-378: Why not use the analytical solution of Spiegelman and McKenzie (1987) for the solid corner flow?

A: The analytical solution of Spiegelman and McKenzie (1987) is derived for a constant mantle viscosity. Because the model for benchmarking will be used for modeling of subduction zones where a temperature and strain-rate dependent viscosity is considered, we decided not to use the analytical solution of Spiegelman and McKenzie (1987). The numerical solution of the analytical corner flow using COMSOL has been successfully benchmarked, we added a new reference (Yu and Lee, 2018) (line 396).

15. Line 382: I don't see an equation 13. The equation numbers jump from 12 to 14.

A: The equation numbering was incorrect. After the correction, the relevant equation is 16 (line 266).

16. Line 383: What do you mean by the uppermost subducting slab? Do you mean the top 2 km of the subducting slab? "Uppermost subducting slab" sounds as if there are more than one subducting slab and the source term is in the top one.

A: The reviewer's understanding is correct. For clarity, we rephrased the sentence as 'uppermost slab-layer' to keep consistency (line 401).

17. Line 424-425: It might be worth discussing here why the relative error increases with model resolution for the uncapped cases.

A: The original figures show that the negative values of the relative error increase with increasing resolution, thus the absolute value of the relative error do decrease. The improved relative error is because of the more accurate calculation of the outflux at the boundaries. However, in the uncapped experiments, the intrinsic errors due to the enforced positive porosity cannot be removed regardless of mesh resolution. We rephrased Section 4.5 accordingly (lines 434-449).

Reviewer #3 (Chenyu Tian)

In this paper, the authors did a great job discussing imposing capped porosity in numerical simulation to avoid unphysical negative situations. I have some comments as follows.

1. The authors indicated using direct fully-coupled PARDISO solver. Can you explain more on the parallelism and the order of accuracy of time integration scheme? Can you include cpu time used with different mesh sizes?

A: We used the fully-coupled PARDISO solver for Eqs. 1–3 and segregated PARDISO solver for Eqs. 4 and 5. The segregated solver allows us to use the Lower Limit option, which is required to enforce the positive porosity in the capped experiments. The generalized-alpha method adopts the second-order accuracy of time integration. We rephrase the sentences in the lines 137-139 and 146-148.
Our study focuses on evaluating the effects of an enforced positive porosity on liquid flow and mass conservation, and we did not track the computational performance. Thus, we do not describe the cpu time.

2. The presentation of figure 2 is not good. It is hard to differentiate lines with similar blues especially (5,3,1) and (7,3,1) lines. The font used in figure 2 is too small.

A: As suggested, we revised Figure 2 by using distinct colors for the triplets (4,2,1), (5,3,1) and (7,3,1), respectively. We also enlarged the font to ease the readability.

3. Figure 8 shows the relative volume-balance error of both capped and uncapped experiments, which indicates that capped porosity produced very large relative error up to 1e3 in some cases. Such large relative error jeopardize the claim made in this paper.

A: In the capped experiments, the large volume-balance error results from the overestimated porosity in the model domain and its subsequent overestimation of the boundary flux: intrinsic error (lines 460-464). Except for this, the distributions of the positive porosity and its peak at the boundary are similar in the capped and uncapped experiments (Sections 4.1-4.4). Thus, our model experiments using the capped porosity can approximate the overall fluid behavior (lines 467-470).

4. The authors claim Figure 8a) shows a significant decrease in the relative volume-balance error. I would suggest run several more smaller mesh size to determine whether the relative error will go negative or not.

A: As described above, we added a finer mesh experiment using an elements size of $\delta_0$/16 to the stagnant porous solid model (Figure 8a).

We are unsure about which experiment the reviewer was referring to. The capped experiment in the model with a stagnant porous solid shows little dependence on mesh resolution and remains positive. The uncapped experiment shows convergence, and our test with the smallest resolution indicates a very small negative relative volume-balance error (about $-2.04*10^{-5}$ %) (lines 435-437). We couldn't run a more refined experiment due to computational expense, and thus we could not check whether the relative volume-balance error became more negative. However given its order of magnitude at a resolution of $\delta_0$/16, we consider that the error is vanishingly small.

5. In this paper, the authors run experiment on four different mesh sizes. I would suggest run more mesh sizes and obtain more data points. The authors can state in the paper if the limitation of computational resources prevent them from using more data points.

A: As discussed above, we additionally conducted simulations using an element size of $\delta_0$/16 which confirmed our former interpretations (lines 428-430): the absolute value of the relative volume-balance error decreases with increasing mesh resolution, particularly for the uncapped experiments. The convergence of the capped experiments is much slower, and the change in its relative error over the variation in the mesh resolution is much smaller than that of what we referred to as the intrinsic error due to capping (see lines 437-439 and 445-448).

6. The authors used a user-defined small porosity for the capped simulation. I would suggest the authors explore different definitions and scaling of the user-defined small porosity. A different definition or scaling of the small porosity might help with decreasing the relative volume balance error.

A: We evaluated the porosity regularization technique described in Section 2, which has been used several times in previous studies (Wilson et al., 2014; Cerpa et al., 2017; 2018). This technique allows us to set the small porosity according to the desired minimum compaction length (described as user-defined) which can be chosen based on the model resolution. Evaluating the other techniques, although useful, is out of the scope of the present manuscript. Instead, we cited Butler (2017) and Sim et al. (2020) to acknowledge other possibilities (lines 122-124).

References

Butler, S. L., Shear-induced porosity bands in a compacting porous medium with damage rheology, Physics of the Earth and Planetary Interiors, 264, 7-17, 2017

Lee, C., A Benchmark for 2-Dimensional Incompressible and Compressible Mantle Convection Using COMSOL Multiphysics ®, Journal of the Geological Society of Korea, 49, 245-265, 2013.

Trim, S. J., Butler, S. L., and Spiteri, R. J.: Benchmarking multiphysics software for mantle convection, Computers & Geosciences, 154, 104797, 2021.

Sim, S. J., Spiegelman, M., Stegman, D. R., and Wilson, C.: The influence of spreading rate and permeability on melt focusing beneath mid-ocean ridges, Physics of the Earth and Planetary Interiors, 304, 106486, 2020.

Yu, S. and Lee, C., A benchmark for two-dimensional numerical subduction modeling using COMSOL Multiphysics®, Journal of the Geological Society of Korea, 54, 683-694, 2018.